# Ocular Surface Workup in Patients with Meibomian Gland Dysfunction Treated with Intense Regulated Pulsed Light

**DOI:** 10.3390/diagnostics9040147

**Published:** 2019-10-13

**Authors:** Luca Vigo, Leonardo Taroni, Federico Bernabei, Marco Pellegrini, Stefano Sebastiani, Andrea Mercanti, Nicola Di Stefano, Vincenzo Scorcia, Francesco Carones, Giuseppe Giannaccare

**Affiliations:** 1Carones Ophthalmology Center, 20122 Milan, Italy; lvigo@carones.com (L.V.); fcarones@carones.com (F.C.); 2Ophthalmology Unit, S.Orsola-Malpighi University Hospital, University of Bologna, 40138 Bologna, Italy; leonardo.taroni1@gmail.com (L.T.); federico.bernabei89@gmail.com (F.B.); marco.pellegrini@hotmail.it (M.P.); 3Ophthalmology Unit, Inferni Hospital, 47923 Rimini, Italy; stefano.sebastiani3@gmail.com (S.S.); andreamercanti.vr@gmail.com (A.M.); nicdist@yahoo.com (N.D.S.); 4Department of Ophthalmology, University Magna Graecia of Catanzaro, 88100 Catanzaro, Italy; vscorcia@libero.it

**Keywords:** intense pulsed light, meibomian gland disease, non-invasive break-up time, dry eye disease, evaporative dry eye

## Abstract

The purpose of the present study was to evaluate changes of signs and symptoms in patients with meibomian gland dysfunction (MGD) treated with intense regulated pulsed light (IRPL), and to further investigate which parameter could predict positive outcomes of the procedure. Twenty-eight patients who bilaterally received three IRPL sessions at day 1, 15, and 45 satisfied the criteria and were included in the study. Non-invasive break-up time (NIBUT), lipid layer thickness (LLT), meibography, tear osmolarity, and ocular discomfort symptoms were measured before and 30 days after the last IRPL session. Qualified or complete success was defined in the presence of an improvement of symptoms associated with an increase of NIBUT (< or ≥ 20%). After IRPL treatment, median NIBUT and LLT increased from 7.5 to 10.2 s and 2.0 to 3.0, respectively (*p* <0.001); tear osmolarity decreased from 304.0 to 301.0 mOsm/L (*p* = 0.002). Subjective symptoms improved after IRPL in 26 patients. Qualified success was reached in 34 eyes, while complete success in 16 eyes. Patients with lower baseline break-up time (BUT) values showed better response to treatment (*p* = 0.04). In conclusion, IRPL improved signs and symptoms in MGD patients, while lower baseline NIBUT values were predictive of better response to IRPL.

## 1. Introduction

Dry eye symptoms are among the most common complaints at ophthalmic practices, impairing patient quality of life and restricting daily activities and work productivity [1,2]. Although various ocular and systemic conditions can determine the onset of dry eye [3,4,5,6], the vast majority of cases originates from a deficiency of the meibomian glands (meibomian gland dysfunction, MGD), that is characterized by a chronic and diffuse abnormality of glands with obstruction of terminal duct and qualitative/quantitative changes of glandular secretion [7,8].

The pathogenesis of MGD is arranged in a vicious circle: meibomian gland inflammation or blockage for ductal epithelium hyperkeratinization leads to stasis of the meibum inside the glands. The reduced gland outflow promotes the proliferation of bacteria, increasing the viscosity of the meibum and thus resulting in further blockage of the gland orifices [9]. Most of the treatments currently available are mainly palliative and consist of hygiene measures and tear substitutes. Antibiotics, anti-inflammatory drugs, and immunosuppressant agents are also used with the aim of breaking the vicious circle. However, therapy often provides only short-term and partial relief of symptoms and signs, with compliance issues.

Intense pulsed light (IPL) has been used in dermatology for over a decade for the treatment of rosacea, acne, and various skin lesions (e.g., benign cavernous hemangioma and telangiectasia) [10,11]. IPL consists of a non-coherent and polychromatic light source with a wavelength spectrum of 500–1200 nm, which can be easily modulated through a proper filter and, when directed to the skin, it is absorbed by chromophores and converted into heat, inducing the ablation of blood vessels.

A new-generation of device (E>Eye), designed specifically for periocular application with calibrated and sculpted sequenced light pulses delivered under the shape of regulated train pulses (intense regulated pulsed light, IRPL), has recently become commercially available, and is currently the only medically certified device for treating dry eye owing to MGD [12,13,14,15,16]. During treatment, the protection of the patient’s eye is mandatory and is obtained thanks to the use of protective shields.

Despite recent studies showed both safety and efficacy of IRPL, predictive parameters of patients who will most likely benefit from the treatment have not yet been identified.

The purpose of this study was to evaluate the changes of a comprehensive ocular surface workup, based on both automated objective and subjective parameters, after IRPL treatment in patients with MGD; in addition, ocular surface parameters predictive of positive outcomes of the procedure were also investigated.

## 2. Materials and Methods

### 2.1. Materials and Patients

This prospective clinical study was conducted between September 2016 and September 2017 at Carones Ophthalmology Center (Milan, Italy). The study was approved by the local Institutional Review Board (approval date 16 May 2016) and was carried out in accordance with the principles of the Declaration of Helsinki. All participants provided written informed consent before any study procedure. Inclusion criteria were age older than 18 years; clinical signs and subjective symptoms of MGD [17]; willingness to continue ongoing therapy (unpreserved tear substitutes, eyelid hygiene); and Fitzpatrick skin scale from 1 to 4. Exclusion criteria were any ocular surface disease other than MGD (e.g., Sjögren’s syndrome, graft-versus-host disease, allergy); previous IPL and/or vectored thermal pulsation therapies within the past 24 months; previous ocular surgery or eyelid trauma; hypotensive eye drops use; punctal plugs; skin pigmented lesions in the treatment area; pregnancy and breastfeeding; any uncontrolled ocular or systemic disease.

### 2.2. Ocular Surface Workup

Ocular surface workup was performed before and 30 days after the last IRPL treatment and included the following automated quantitative measurements: non-invasive break-up time (NIBUT), lipid layer thickness (LLT), non-contact infrared meibography, and tear osmolarity. The Ocular Surface Disease Index (OSDI) questionnaire was administered before any procedure to score ocular discomfort symptoms. In addition, patient’s satisfaction after treatment was ascertained by asking the patients whether they perceived improvements from their baseline symptoms according to a 5-grade scale: none = 0, trace = 1, mild = 2, moderate = 3, high = 4.

The I.C.P. Tearscope (SBM Sistemi, Turin, Italy), a lighting system that allows the in vivo visualization of the different layers of the tear film at the slit lamp under magnification × 16–25, was used to measure NIBUT and LLT [18]. The median value of three successive measurements was used for statistical analysis. Lipid layer patterns were classified based on their appearance, and thickness was graded from 0 to 5: absence of lipids (grade 0); open meshwork (grade 1); tight meshwork (grade 2); waves (grade 3); amorphous (grade 4); and color mixing (grade 5) [12,19]. Tear osmolarity was evaluated with TearLab Osmolarity System (TearLab Corporation, San Diego, CA, USA) by obtaining a sample from the lower temporal eyelid tear film meniscus [20].

Non-contact infrared meibography was performed by using the I.C.P. MGD meibography system (SBM Sistemi, Turin, Italy) to acquire infrared images of the meibomian glands after everting the lower eyelid [21,22]. Meibomian gland loss (MGL) was defined as the percentage of gland loss in relation to the total tarsal area of the lid. Images were digitally analyzed and MGL was measured using I.C.P. application.

### 2.3. Treatment Procedure 

IRPL treatments were performed using E>Eye device (E-Swin, Paris, France), set on the proprietary “dry eye mode”. Treatment intensity was determined based on patient’s Fitzpatrick skin type, ranging from 9.8 to 13 J/cm^2^. Patients received three treatment sessions performed at day 1, 15, and 45, as per manufacturer recommendations [12]. During each treatment, protective eye shields were placed over the eyes and ultrasound gel was applied to the treatment area. Five flashes were applied for each eye starting from the inner canthus and ending on the temporal region below the lower eyelid, with slight overlapping applications. According to our protocol, all patients continued their ongoing therapy (unpreserved tear substitute and eyelid hygiene); in addition, all patients instilled 0.3% cortisol phosphate in hyaluronic acid vehicle eye drops (Cortivis; Medivis, Catania, Italy) twice daily for 10 days after the first session of IRPL.

### 2.4. Main Outcome Measures

The primary outcomes were the changes of each ocular surface parameter analyzed 30 days after the last IRPL session (at day 75), and the rates of qualified and complete success. In detail, qualified success was defined in the presence of an improvement of symptoms (score ≥2) associated with an increase of NIBUT; complete success was defined in the presence of an improvement of symptoms (score ≥3) associated with an increase of NIBUT ≥20%.

The secondary outcome was the detection of ocular surface parameters predictive of the success of the procedure.

### 2.5. Statistical Analysis

The SPSS statistical software (SPSS Inc Version 22.0, Chicago, IL, USA) was used for data analysis. Values are expressed as median, interquartile range (IQR), and 95% confidence intervals (CIs). The difference between pre- and post-treatment values for each parameter was calculated and reported as delta (Δ). The non-parametric Wilcoxon test was used to compare variables before and after IRPL treatment. The Mann–Whitney U test was used to compare variables between patients who experienced success after the procedure and those who did not. Spearman’s correlation was run to determine the relationships between variables. A *p* <0.05 was considered statistically significant.

## 3. Results

Eighty eyes of 40 patients underwent IRPL treatment during the study period. Of these, 56 eyes of 28 patients (6 males, 22 females; median age 46.0 years, IQR: 17.5) received regular treatment sessions and follow-up visits and were finally included in the analysis. The remaining patients were lost to follow-up (four patients) or did not complete the whole treatment cycle (eight patients). Demographic characteristics and ocular surface parameters at baseline of the included patients are reported in Table 1.

Thirty days after the last IRPL session, median NIBUT significantly increased from 7.5 s (IQR: 5.2, 95% CI: 7.0–8.5) to 10.2 s (IQR: 5.4, 95% CI: 9.5–11.4) (*p* <0.001; Figure 1, part A), and median LLT grade significantly improved from 2.0 (IQR: 1.0, 95% CI: 1.5–1.9) to 3.0 (IQR: 2.0, 95% CI: 2.6–3.3) (*p* <0.001, Figure 1, part B). No statistically significant changes were recorded after IRPL treatment for both MGL (Figure 1, part C) and OSDI score (always *p* >0.05). Median tear osmolarity significantly decreased after treatment from 304.0 mOsm/L (IQR: 9.8, 95% CI: 302.9–308.3) to 301.0 mOsm/L (IQR: 14.0, 95% CI: 298.1–303.2) (*p* = 0.002, Figure 1, part D). Twenty-six patients (92.9% of the total) showed an improvement of symptoms after treatment (median grade 2.0 (IQR: 2.0, 95% CI: 2.0–2.5)). Figure 2 shows the distribution of patients’ perceived improvement in symptoms according to the 5-grade scale used.

Qualified success (improvement of both symptoms and NIBUT) was reached in 34 eyes (60.7% of the total), while complete success (improvement of symptoms (score ≥3) associated with an increase of NIBUT (≥20%)) in 16 eyes (28.6% of the total). Patients who achieved qualified success had significantly lower NIBUT values at baseline compared to the others (respectively 6.7 s (IQR: 3.9, 95% CI:5.4–7.7) vs. 8.7 s (IQR: 5.3, 95% CI:7.5–10.1)) (Figure 3). Conversely, no differences were found at baseline between patients belonging to the success group vs. others for LLT, meibomian gland loss, and tear osmolarity (always *p* >0.05). A significant negative correlation was found between baseline NIBUT and Δ NIBUT (*p* <0.001; R = −0.463), baseline osmolarity and Δ osmolarity (*p* <0.001; R = −0.584), and baseline meibomian gland loss and Δ meibomian gland loss (*p* <0.001; R = −0.470) (Figure 4).

No adverse effects related to the treatment were reported at any visit of the study.

## 4. Discussion

Meibomian gland dysfunction has been identified as the most common cause of dry eye disease [8]. Treatments currently available are mainly palliative solutions, often insufficient to improve clinical signs and overcome patient’s discomfort symptoms. Intense pulsed light was recently introduced in the field of ophthalmology and recent clinical studies showed that it is able to provide an improvement of both signs and symptoms in MGD patients (Table 2) [12,13,14,15,16,23,24,25,26,27,28,29,30,31,32,33,34,35]. IPL can be performed in combination with other therapies, like meibomian gland expression, and thus also represents a promising complementary treatment for MGD. [14,24,26,30,31,35]. This combination of treatments allowed for the successful management of refractory cases of MGD, as demonstrated in a recent multicenter prospective study [35]. It was recently demonstrated that the treatment reduces consistently also tear inflammatory cytokines [33,34]. Although different speculative pathophysiological theories have been proposed to explain the positive effects of intense pulsed light upon dry eye signs and symptoms, the mechanisms of actions are still not fully elucidated. Among these, the coagulation of superficial blood vessels and telangiectasias of eyelids skin induced by light energy, the heating and liquefying of meibomian glands secretions with improved viscosity and outflow, and the decrease of bacterial and parasitic load over eyelids and eyelashes have been proposed [36]. More recently, the enhancement in collagen synthesis and connective tissue remodeling, the reduction in skin epithelial cell turnover, and the modulation of cellular inflammatory markers have also been hypothesized [37].

To date, most of the available studies reported improvements in terms of lid margin features (e.g., thickening and vascularity, telangiectasia, number of plugged glands) and meibomian gland secretion quality and expressibility [13,14,25,26,28,31]. However, these measures are subjective, and prone to observer bias due to a low degree of standardization. Conversely, a comprehensive ocular surface workup with automated quantitative measurements may overcome these drawbacks, thus improving the objective monitoring of the disease course after treatment [38].

In the present study, NIBUT, LLT, non-contact meibography, and tear osmolarity have been investigated before and after IRPL sessions. Our results confirmed that NIBUT significantly increases after IRPL [12,13,14,15,16,23,25,26,27,29,30,31,32,35]. Since the role of tear film instability as pivotal mechanism of DED onset and persistence has been gaining prominence [39,40], its significant improvement after IRPL therapy represents a major goal of the procedure.

Only few previous studies evaluated LLT changes after IPL, with conflicting results [12,26,29]. Despite the lack of universal consensus on the LLT modifications after IPL therapy, Ahmed and co-authors demonstrated an improvement in lipid content and composition (and in particular of polar lipids that critically impact the health of the ocular surface), contributing to the stability of the tear film [41]. In the present study, we showed a significant increase of LLT by using the same grading scale previously utilized by Craig and collaborators [12].

Tear osmolarity significantly decreased after treatment, in contrast with previous studies which conversely reported no changes of this parameter after treatment [12,14,26]. Since tear osmolarity has been shown by the TFOS Dry Eye Workshop II to be influenced by both tear film stability and lipid layer characteristics [41], the improvement of these two parameters after IRPL could explain this finding. However, it should be pointed out that in our study, as well as in other MGD populations, tear osmolarity values were within the upper range of normality [14,20,42].

In the present study, the area of MGL did not change after IRPL treatment. This finding is in contrast with the only study evaluating before this parameter, which reported a 5% decrease of MGL after IPL in treatment-naïve patients [27]. However, this finding is controversial since a similar improvement was noted in the same study also in control patients treated with eyelid hygiene alone.

Subjective symptoms were investigated in the present study by both OSDI score and a five-grade scale specifically focused on patients’ perceived improvement in symptoms after treatment. Despite the lack of significant decrease of OSDI score after treatment, the vast majority of patients reported an overall improvement of discomfort symptoms, which was classified as moderate to high in almost half of the patients. This amelioration of subjective symptoms is in agreement with previous studies, which employed both validated questionnaires and specific scales of satisfaction [12,13,14,23,24,25,26,32,33,35].

As several options are available in the therapeutic algorithm of MGD, studies aiming at predicting patients who will most likely benefit from IRPL are desirable in order to guide the selection of the best patient population for this type of treatment. Therefore, we tried to identify which factors might predict a positive outcome of IRPL by analyzing the characteristics of patients before treatment. We found that patients with lower values of NIBUT at baseline showed a better response after IRPL treatment. On the other hand, none of the other parameters evaluated at baseline were able to discriminate patients who experienced the most benefit from treatment. In addition, significant negative correlations between each parameter and its variation after IRPL treatment were found, except for LLT. Therefore, those patients presenting with worse baseline value of one parameter experienced a greater improvement of the parameter after treatment.

The major limitation of the study is represented by the lack of a control group, which makes it difficult to rule out whether improvements could be related specifically to the IRPL treatment, the natural fluctuations of the disease course, or the placebo effect. Additionally, the relatively small size of the population might hamper detection of further significance in case of small differences among parameters. For this reason, we included both eyes of each patient in the final analysis, and this may represent a bias. However, since the intraclass correlation coefficients between right and left eyes were high, especially for LLT (0.876) and MGL (0.765) but also for tear osmolarity (0.634) and NIBUT (0.542), further studies with larger sample sizes including only one eye for each patient are desirable in order to confirm these findings.

In conclusion, IRPL for the treatment of patients with dry eye owing to MGD improved NIBUT, LLT, and tear osmolarity, as well as subjective symptoms. Patients with a higher tear film instability at baseline responded better to the procedure and would likely represent the ideal candidates for this treatment.

## Figures and Tables

**Figure 1 diagnostics-09-00147-f001:**
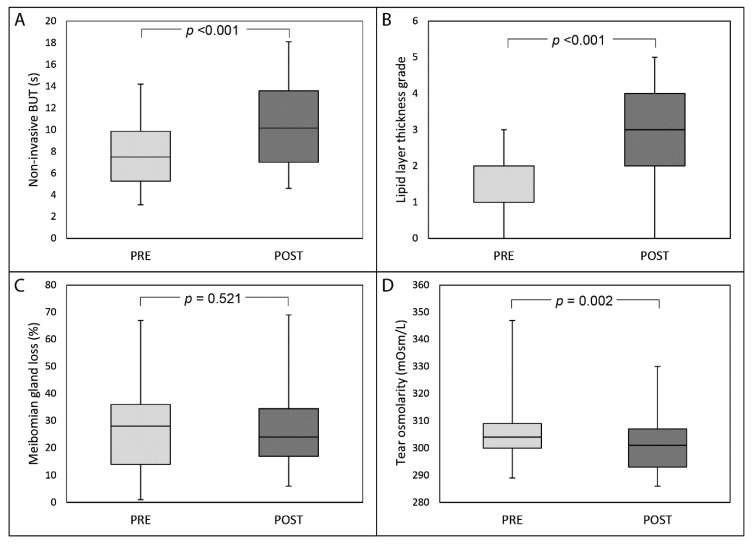
Box-plot analysis of non-invasive break-up time (NIBUT) (part **A**), lipid layer thickness grade (part **B**), meibomian gland loss (part **C**) and tear osmolarity (part **D**) before and 30 days after the last session of intense regulated pulsed light treatment.

**Figure 2 diagnostics-09-00147-f002:**
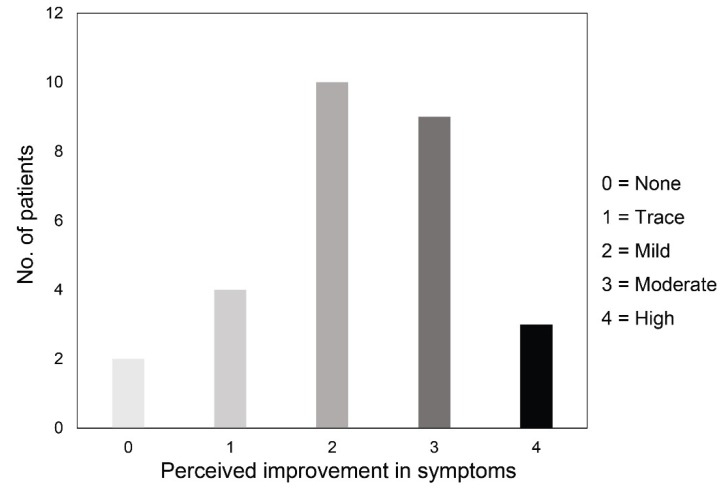
Distribution of patients according to the 5-grade scale about their perceived improvement in symptoms after intense regulated pulsed light treatment.

**Figure 3 diagnostics-09-00147-f003:**
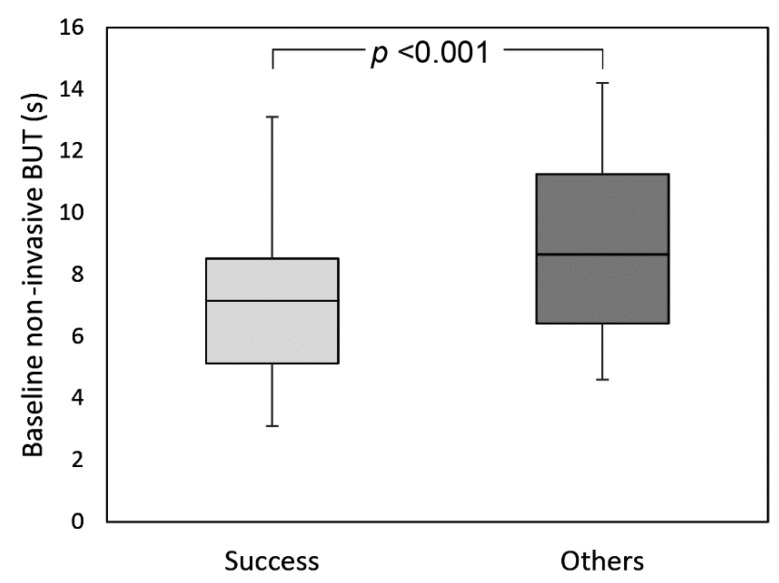
Box-plot analysis of baseline non-invasive break-up time in patients belonging to the success group and in the others.

**Figure 4 diagnostics-09-00147-f004:**
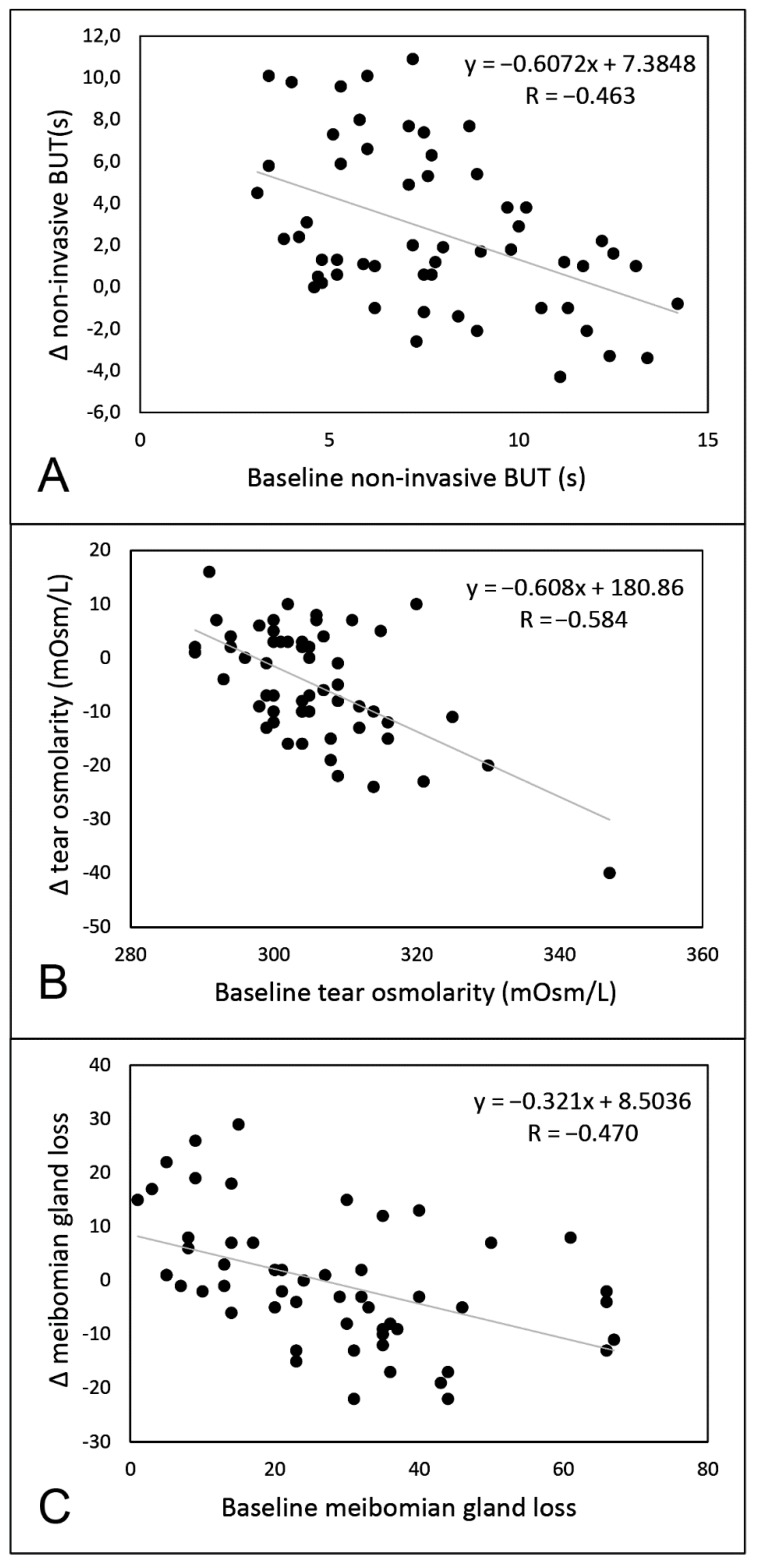
Linear regression analysis between baseline value and delta (Δ) value for non-invasive break-up time (part **A**), tear osmolarity (part **B**), and meibomian gland loss (part **C**).

**Table 1 diagnostics-09-00147-t001:** Demographic and clinical characteristics of patients at baseline.

Characteristic	Number (IQR)	95% CI
Patients (Eyes)	28 (56)	-
Male/female ratio	0.27	-
Age (years)	46.0 (17.5)	44.6–51.9
Non-invasive BUT (s)	7.5 (5.2)	7.0–8.5
Lipid layer thickness (grade)	2.0 (1)	1.5–1.9
Meibomian gland loss (%)	28.0 (22.8)	23.1–32.6
Tear osmolarity (mOsm/L)	304.0 (9.8)	302.9–308.3
OSDI (score)	25.0 (34.3)	23.2–36.4

Values are expressed as median (IQR). IQR: interquartile range; CI: confidence interval; BUT: break-up time; OSDI: Ocular Surface Disease Index.

**Table 2 diagnostics-09-00147-t002:** Clinical studies about the use of intense pulsed light treatment for meibomian gland dysfunction.

Study	Device	No. Sessions	No. Patients/Eyes	Effects on Symptoms	Effects on Signs
Craig et al 2015. [12]	E>Eye (E-Swin)	3	28/28	SPEED decrease	Improvement of NIBUT and LLT; no change of TER, TMH, and osmolarity
Toyos et al 2015. [23]	Q4 (Dermamed)	7	78/156	/	Improvement of TBUT
Vegunta et al 2016. [24]	Q4 (Dermamed)	4	35/70	SPEED2 decrease	Improvement of MGE
Jiang et al 2016. [13]	E>Eye (E-Swin)	4	40/40	Significant improvement	Improvement of BUT, MGE, lid margin signs; no change of TMH
Gupta et al 2016. [25]	Q4 (Dermamed)	4	100/200	OSDI decrease	Improvement of BUT, MGE and lid margin signs
Albietz et al 2018. [14]	E>Eye (E-Swin)	3	26/52	OSDI decrease	Improvement of BUT, MGE, CFS and lid margin signs; no change of Schirmer, osmolarity, corneal sensitivity, lid margin bacteria colonies
Dell et al 2017. [26]	M22 (Lumenis)	4	40/80	SPEED decrease	Improvement of BUT, MGE, CFS; no change of osmolarity and LLT
Guilloto et al 2017. [15]	E>Eye (E-Swin)	4	36/72	/	Improvement of BUT, TMH and Schirmer
Yin et al 2017. [27]	M22 (Lumenis)	3	18/18	OSDI decrease	Improvement of BUT, MG expressibility, dropout and microstructure
Rong et al 2018. [30]	M22 (Lumenis)	3	28/28	SPEED decrease	Improved MG secretion function and TBUT
Seo et al 2018. [28]	M22 (Lumenis)	3	17/34	OSDI decrease	Improvements in the lower lid margin vascularity, meibum expressibility and quality
Arita et al 2018. [35]	M22 (Lumenis)	4 to 8	31/62	SPEED decrease	Significant improvement of NIBUT, BUT, tear interferometric fringe grading, meibum grade, lid margin abnormality scores, CFS
Arita et al 2019. [29]	M22 (Lumenis)	8	22/44	SPEED decrease	Significant improvement of lipid layer grade, LLT, NIBUT, BUT, lid margin abnormalities, and meibum grade, CFS
Mejia et al 2019. [31]	E>Eye (E-swin)	3	25/50	/	Improved symptoms, Shirmer test, TBUT, VB score
Li et al 2019. [32]	M22 (Lumenis)	3	40/80	OSDI decrease	TBUT improvement
Vigo et al 2019. [16]	E>Eye (E-Swin)	3	19/38	Improvement	NIBUT and LLT increase

BUT: break-up time; CFS: corneal fluorescent staining; LLT: lipid layer thickness; MGE: meibomian gland evaluation; NIBUT: non-invasive break-up time; OSDI: Ocular Surface Disease Index; SPEED: Standard Patient Evaluation of Eye Dryness; TER: tear evaporation rate; TMH: tear meniscus height.

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
