# Peer review of "Ocular Surface Workup in Patients with Meibomian Gland Dysfunction Treated with Intense Regulated Pulsed Light"

_diagnostics, 2019, doi:10.3390/diagnostics9040147_

Round 1
Reviewer 1 Report
Intense regulated pulsed light (IRPL) for the treatment of patients with dry eye owing to MGD improved NIBUT. IRPL improved signs and symptoms in MGD patients. Lower baseline NIBUT values were predictive of better response to IRPL. These results may provide an option of tretment in MGD patients.Author Response
- Intense regulated pulsed light (IRPL) for the treatment of patients with dry eye owing to MGD improved NIBUT. IRPL improved signs and symptoms in MGD patients. Lower baseline NIBUT values were predictive of better response to IRPL. These results may provide an option of treatment in MGD patients.
Reply: Dear Reviewer, thanks for your positive comments. Your comment summarizes perfectly the results of our paper.
Reviewer 2 Report
The authors said "Patients received 3 treatment sessions performed at day 1, 15 and 45" and reported the results 30 days after treatment.
Many IPL studies have been conducted.
RCT studies are needed to validate their effectiveness.
Author Response
Dear Reviewer, thanks for your comments.
- The authors said "Patients received 3 treatment sessions performed at day 1, 15 and 45" and reported the results 30 days after treatment.
Reply: We performed IPL procedure at day 1, 15, 45 and then we repeated the ocular surface workup 30 days after the last IPL treatment (at day 75). For improving clarity, we added "(at day 75) also in the text.
- Many IPL studies have been conducted.
Reply: We agree with you that despite IPL procedure is of recent introduction in the Ophthalmic field, several clinical studies have been already published. Therefore, we created the table n.2 in order to summarize extensively data from these studies. However, to our knowledge the identification of ocular parameters predictive of a better response to IPL treatment, attempted in the present paper, is not addressed in previous studies.
- RCT studies are needed to validate their effectiveness.
Reply: We agree with you that there is the need of further RCTs to validate the effectiveness of the procedure as clearly stated in the last sentence of the discussion: "Randomized clinical trials are needed to validate the effectiveness of the procedure."
Reviewer 3 Report
Dry eye disease is defined as a multifactorial disease of the ocular surface characterized by a loss of homeostasis of the tear film and accompanied by ocular symptoms that result in part from tear film instability and hyperosmolarity, ocular surface inflammation and damage, and neurosensory abnormalities. In order to make the paper more complete it wouldbe good to briefly mention the content of the interesting articles below. Therefore, I suggest the authors to study the important recently published articles, incorporate their meaning and briefly mention them in the text and in the list of references. Multicenter Study of Intense Pulsed Light Therapy for Patients With Refractory Meibomian Gland Dysfunction.
Arita R, Mizoguchi T, Fukuoka S, Morishige N. Cornea. 2018:1566-1571.
To make this article less arid and therefore, more interesting, there is a need for a small introduction regarding eye protection.
At line 78 the authors write "Qualified success was reached in 34 eyes (60.7% of the total), while complete success in 16 eyes (28.6% of the total). Patients who achieved qualified success had significantly lower NIBUT values at baseline compared to the others (respectively 6.7 s [95% CI:5.4-7.7] vs 8.7 s [95% CI:7.5-10.1]) (Figure 3)." . It is not immediately obvious how this conclusion was arrived at and also not obvious what is meant by "success" and "the others". It would be helpful if the authors could rephrase and expand upon this sentence.
Author Response
- Dry eye disease is defined as a multifactorial disease of the ocular surface characterized by a loss of homeostasis of the tear film and accompanied by ocular symptoms that result in part from tear film instability and hyperosmolarity, ocular surface inflammation and damage, and neurosensory abnormalities. In order to make the paper more complete it wouldbe good to briefly mention the content of the interesting articles below. Therefore, I suggest the authors to study the important recently published articles, incorporate their meaning and briefly mention them in the text and in the list of references. Multicenter Study of Intense Pulsed Light Therapy for Patients With Refractory Meibomian Gland Dysfunction.
Arita R, Mizoguchi T, Fukuoka S, Morishige N. Cornea. 2018:1566-1571.
Reply: Dear Reviewer, thanks for the suggestion. We cited the article from Arita and collaborators (Cornea 2018) in the table n.2, in the discussion and in the reference list (n. 28). We added in the discussion the following sentences:
"IPL can be performed in combination with other therapies, like meibomian gland expression, and thus represents also a promising complementary treatment for MGD. [13,17,19,23,24,28]. This combination of treatments allowed to manage successfully also refractory cases of MGD, as demonstrated in a recent multicenter prospective study [28].
- To make this article less arid and therefore, more interesting, there is a need for a small introduction regarding eye protection.
Reply: We addded in the intro the following sentence that is reported also in the methods section: "During treatment, the protection of the patient’s eye is mandatory and is obtained thanks to the use of protective shields."
At line 78 the authors write "Qualified success was reached in 34 eyes (60.7% of the total), while complete success in 16 eyes (28.6% of the total). Patients who achieved qualified success had significantly lower NIBUT values at baseline compared to the others (respectively 6.7 s [95% CI:5.4-7.7] vs 8.7 s [95% CI:7.5-10.1]) (Figure 3)." . It is not immediately obvious how this conclusion was arrived at and also not obvious what is meant by "success" and "the others". It would be helpful if the authors could rephrase and expand upon this sentence.
Reply: We agree with you that it is not ease for the reader to understand the concept of "qualified" or "complete" success. The requested format of the article (materials and methods at the end of the article) does not help. Therefore, we specified the details of qualified and complete success not only in the methods section (as already done in the previous version) also in the results section, as follows:
"Qualified success (improvement of both symptoms and NIBUT) was reached in 34 eyes (60.7% of the total), while complete success (improvement of symptoms [score ≥ 3] associated with an increase of NIBUT [≥ 20%]) in 16 eyes (28.6% of the total)."